# Research on the Gut Microbiota of Hainan Black Goat

**DOI:** 10.3390/ani12223129

**Published:** 2022-11-13

**Authors:** Wenbo Zhi, Kai Tang, Jinsong Yang, Tianshu Yang, Rong Chen, Jiaming Huang, Haisheng Tan, Jianguo Zhao, Zhanwu Sheng

**Affiliations:** 1College of Food Science and Engineering, Hainan University, Haikou 570228, China; 2College of Materials Science and Engineering, Hainan University, Haikou 570228, China; 3Laboratory of Tropical Veterinary Medicine and Vector Biology, School of Life Sciences, Hainan University, Haikou 570228, China; 4Haikou Experimental Station, Chinese Academy of Tropical Agricultural Sciences, Haikou 571101, China

**Keywords:** Hainan black goat, gut microbiota, microbial diversity, high throughput sequencing

## Abstract

**Simple Summary:**

The homeostasis of intestinal microorganisms has an important effect on the healthy development of the host, as well as for ruminants. However, the study on the intestinal microflora of the Hainan Black Goat, a special breed in Hainan, is not sufficient. Therefore, this study is devoted to the preliminary study of the composition of gut microbiota and the differences between the different intestinal segments of Hainan black goats. According to the results, the large intestine has a greater variety and a greater richness of the intestinal microbiota community than the small intestine. The microbial composition of the cecum and colon is similar. The results of this study contribute to our understanding of the species diversity of gut microbiota in different intestinal segments of Hainan black goats.

**Abstract:**

The intestine of animals is a complex micro-ecosystem containing a large number of microbiomes, which is essential for the host’s health development. The Hainan black goat with good resistance and adaptability is a unique species in Hainan, China. These unique physiological characteristics are inseparable from their intestinal microbiota. In this study, high-throughput sequencing was used to investigate bacterial communities in different segments of the intestinal tract of Hainan black goat. The results showed that the indices of Chao1 and ACE in the cecum and colon were significantly greater than those in the ileum (*p* = 0.007, 0.018). According to PCoA, the intestinal flora composition of the cecum and colon is almost equivalent. In contexts of the phylum, *Firmicutes, Bacteroidota*, and *Pseudomonadota* were the dominant phyla in the gut of the Hainan black goat. While in context of the genus, the dominant groups in the gut of black goats mainly include *Ruminococcaceae_UCG-005, Bacteroides, Paeniclostridium, Christensenellaceae_R-7_group, Rikenellaceae_RC9_gut_group,* and *Eubacterium coprostanoligenes _group, Prevotella_1*, they have different proportions in different intestinal segments. The gut microbiota of Hainan black goat is mainly *Firmicutes, Bacteroidota*, and *Pseudomonadota*. Influenced by the intestinal location where they colonize, the large intestine has a more complex intestinal flora than the small intestine. In contrast, there are only minor differences between the caecum and the colon in the large intestine.

## 1. Introduction

Mammal gastrointestinal systems are home to numerous, intricate microbes. These microbes affect the digestive system, physiological system, and immune system of the host [1,2,3]. Some mammals such as ruminants have different microbial structures due to physiology and digestion modes [4]. Numerous elements, including but not restricted to food, geography, and genetics, have an impact on the community structure of the intestinal microflora [5,6,7,8]. Studies have demonstrated that gut microbiota can affect the host’s health and immune system [9]. However, except for the rumen and feces, there are only few studies on the microflora of the large intestine and small intestine of ruminants [10,11,12].

Hainan black goats are mainly active in Hainan, China, with good resistance and adaptability [13]. Hainan black goat is mainly used for meat. The meat odor lacks a urine smell, the fat composition is uniform and not greasy, and has high nutritional value. Hainan black goat meat is regarded as one of the “four famous foods” in Hainan [14,15]. The meat protein content is more than 22.6% and is rich in most essential amino acids, while fat and cholesterol content are less than 3% and 60 mg/kg. According to statistics, there were approximately 1,371,000 Hainan black goats in Hainan, China in 2017.

High-throughput sequencing (HTS) include 16S, 18S, ITSrDNA-based amplicon sequencing, macro-genome sequencing, whole genome sequencing [16,17]. With the development of HTS, the cost required has become lower, the accuracy of sequencing results has improved, and these have made it become a reality to research microbial communities in such a complex environment [18]. The dynamics of the gastrointestinal bacterial community in goats at different developmental stages were studied using HTS, revealing that the microbial richness and diversity increased as animals aged [19]. A recently published study explored information on the distribution of intestinal flora in Yimeng black goats [20]. Compared to Yimeng, Hainan has a typical tropical climate. Therefore, the study of gut microorganism of Hainan black goats is more conducive to reveal the relationship with climate and animal physiology and indirectly with microbiomes.

In this study, we conducted a zonation study and functional analysis of microbial communities colonizing different intestinal segments of Hainan black goats by HTS. We hope that our study will help to reveal the intestinal microbial structure of Hainan black goats and thus identify the differences from other goats.

## 2. Materials and Methods

### 2.1. Materials and Grouping Status

The experimental animals were reared and maintained in Chengmai County, Hainan Province. Three healthy one-year-old Hainan black goats (31.05 ± 5.51 kg body weight) were used in this study. The feed composition is mainly king grass and peanut stalk silage. The animals were slaughtered under the supervision of the Hainan University Institutional Animal Use and Care Committee (Permit No. GB/T 35892-2018). During slaughtering, the two ends of the intestines were knotted with cotton rope, and the intestines were brought back to the laboratory on ice within 2 h. Five groups of samples were taken from each intestine, namely group D (duodenum), group J (jejunum), group I (ileum), group CA (caecum), and group CO (colon). Additionally, the same amount of fresh intestinal contents were collected and mixed in the anterior, middle, and posterior parts of each intestine with a cryopreservation tube, and stored at −80 °C until further analysis. The above operations were carried out in a sterile environment.

### 2.2. DNA Extraction

The DNA extraction method adopted by Li, A. was followed [20].

### 2.3. PCR Amplification and Miseq Sequencing

Take 30 ng extracted DNA sample for PCR. The common primers of V3/V4 and V4/V5 regions are commonly used for microbial analysis, and they are sensitive to intestinal microorganisms and marine archaea, respectively [21,22]. Therefore, V3/V4 primers (341F:ACTCCTACGGGAGGCAGCAG; 806R:GGACTACHVGGGTWTCTAAT) were used for PCR. The concentration of DNA obtained from 2% gel sugar extraction with the kit (Axygen, Shanghai, China) was determined using the QuantiFluor™-ST fluorescence quantification system (Promega, San Luis Obispo, CA, USA). PE amplicon library construction was performed on Illumina Miseq platform (Shanghai, China).

### 2.4. Bioinformatics and Statistical Analysis

Mothur (v.1.39.1) was used to filter, splice, and remove the problematic sequences to screen effective tags. OTUs were obtained by clustering effective tag sequences with Uparse (v9.2.64_i86linux32) (similarity ≥ 97%). According to the OTUs sequence information and abundance information, various analyses, such as species annotation, community diversity, and intergroup differences, were carried out one by one.

QIIME (Version 1.9.1) was used to analyze alpha diversity and beta diversity, and used R software (3.2.1, CRAN, Vienna, Austria) to plot, including the rank abundance and rarefaction curves, PCoA, and Venn diagram. The community diversity and community richness of the sample are described by Chao1, Shannon, Simpson, and ACE indices. Principal Co-ordinates Analysis (PCoA) was used to calculate the (un) weighted UniFrac distances to represent the differences between different samples. Biomarker features in each group were screened by Metastats (20090414, https://cbcb.umd.edu/software/metastats, accessed on 1 September 2022) and LEfSe software (v1.0, https://huttenhower.sph.harvard.edu/lefse/, accessed on 1 September 2022). PICRUST (v.1.1.0, http://picrust.github.io/picrust/, accessed on 1 September 2022) was applied to predict the potential molecular functions of the bacterial flora within the samples.

Statistical analysis was performed by SPSS (R26.0.0.0). Each group of samples is repeated 3 times. All data were tested for Tukey test and Kruskal-Walla’s test. The means of the two groups were compared using the *t*-test and Wilcoxon rank-sum test. The difference between the two groups of samples was tested by Metastats software (20090414). *p* (correction with FDR) <0.05: a significant difference. Data form: mean ± standard deviation.

## 3. Results

### 3.1. Analysis of DNA Sequences

An average of 108,334 high-quality reads were recovered with an average sequence length of 368 bp from all samples after taxonomic identification, yielding a total of 1,625,020 high-quality reads. On average, 3416 OTUs were screened per sample (similarity ≥ 97%).

Specifically, OTUs are available in all intestinal segments (6766 OTUs in the duodenum, 4424 OTUs in the jejunum, 1490 OTUs in the ileum, 5789 OTUs in the cecum, and 6497 OTUs in the colon). Six-thousand OTUs were unique for the duodenum, 1694 OTUs were unique for the jejunum, 420 for the ileum, 2222 for the cecum, and 2789 for the colon (Figure 1a). Remarkably, we also observed 199 and 3219 OTUS common to each part of the small intestine and each part of the large intestine, respectively (Figure 1b,c). It is a sign that the conditions for DNA extraction and analysis have been satisfied when the species accumulation curve and rank abundance curve are beginning to stabilize (Figure 1d,e).

### 3.2. Analysis of Microbial Diversity

The diversity and richness of intestinal microbiota in different intestinal segments were analyzed by alpha diversity. Chao1 and ACE indexes were used to describe the community richness of the sample, and Shannon and Simpson indexes were used to describe the community diversity of the sample.

All indexes indicated that the microbial community richness and diversity in the large intestine (cecum, colon) were higher than those in the small intestine (Figure 2a–c). Meanwhile, Chao 1 and ACE indexes in the cecum and colon were significantly larger than those in the ileum (*p* = 0.007, 0.018). Similar to Yimeng black goat [19], the species diversity within the ileum was lower than that found for duodenum and jejunum samples.

Through PCoA, the colony difference between different intestinal segments was described as UniFarc distance for beta diversity analysis. The UniFarc distance revealed the microbiota in the cecum and colon groups were clustered together, indicating a high degree of microbial community similarity. In contrast, the microbiota in the jejunum and ileum groups were more dispersed and further apart from the cecum and colon, indicating a low degree of microbial similarity between the samples.

On the whole, the results of the beta diversity analysis were similar to that of Yimeng black goat, but the details were different. On the one hand, in Figure 3a, the microflora of the cecum and colon are more similar, the microbial ecology of the duodenum and jejunum is more similar, and the ileum, as a transitional intestine, is closer to the duodenum and jejunum. On the other hand, in Figure 3b, the cecum and colon are clustered, the ileum and jejunum are clustered close to the cecal colon, and the duodenum is the farthest.

### 3.3. The Composition of Gut Microbiota

In the context of the phylum, a total of 20 phyla bacterial communities were detected in the intestinal contents of Hainan black goats, the top three gut microbiotas in the duodenum (62.44%, 30.55%, 1.83%), jejunum (75.19%, 11.68%, 1.31%), cecum (77.11%, 18.53%, 1.89%) and colon (66.78%, 26.31%, 2.62%) were Firmicutes, Bacteroidota and Pseudomonadota, and ileum (83.77%, 11.18%, 2.95%) were predominated by Firmicutes, Pseudomonadota and Bacteroidota. Other phyla (Pseudomonadota, Actinobacteria, Saccharibacteria, Spirochetes, Cyanobacteria, Fibrobacteres, and Elusimicrobia) were detected at a lower abundance. 

In the context of the genus, 325 genera were detected. Figure 4 shows the top ten genera in terms of abundance. The most abundant genera detected included: *Paeniclostridium, Ruminococcaceae_UCG-005, Christensenellaceae_R-7_group, Bacteroides, Clostridium_sensu_stricto_1, Rikenellaceae_RC9_gut_group, Eubacterium coprostanoligenes_group, Prevotella_1, Ruminococcaceae_UCG-010, Ruminococcaceae_UCG-014*. The gut microbiota in the duodenum (32.26%, 10.81%, 8.82%) was predominated by *Paeniclostridium, Rikenellaceae_RC9_gut_group,* and *Prevotella_1* in descending order, the gut microbiota in the jejunum (32.99%, 7.27%, 3.68%) was predominated by *Paeniclostridium, Ruminococcaceae_UCG-005 and Christensenellaceae_R-7_group* in descending order, the gut microbiota in the ileum (25.93%, 10.52%, 5.12%) was predominated by *Clostridium_ sensu_stricto_1, Paeniclostridium and Ruminococcaceae_UCG-005* in descending order, the gut microbiota in the cecum (15.06%, 8.51%, 8.10%) was predominated by *Ruminococcaceae_UCG-005, Christensenellaceae_R-7_group,* and *Bacteroides*, the gut microbiota in the colon (11.70%, 8.29%, 8.71%) was predominated by *Ruminococcaceae_UCG-005, Christensenellaceae_R-7_group,* and *Eubacterium coprostanoligenes_group* in descending order.

In addition, the relative abundance of *Ruminococcaceae_UCG-005* was significantly higher in the cecum (15.06%) than in the duodenum (0.16%), jejunum (3.68%) and ileum (5.12%) (*p* = 0.014). The relative abundance of *Bacteroides* was significantly higher in the colon (8.71%) than in the duodenum (0.16%) and ileum (0.63%) (*p* = 0.043). The relative abundance of *Eubacterium coprostanoligenes _group* was significantly more in the cecum (8.10%) than in the duodenum (0.41%) and jejunum (1.41%) (*p* = 0.047). The relative abundance of *Ruminococcaceae_UCG-010* was significantly higher in the cecum (6.61%) than in the duodenum (0.34%) (*p* = 0.021).

Metastats software was used to examine (*p* < 0.05, FDR < 0.1), and LEfSe software was used to draw the difference in microbial abundance among groups. The results of LEfSe showed that there were differences in microbial abundance among groups (Figure 5) Metastats results showed that the abundance of *Ruminococcaceae_UCG-005*, *Ruminiclostridium,* and *Anaerofilum* in the cecum were significantly higher than those in duodenum (*p* = 0, 0.005, 0.001) (Table 1). The abundance of *Bacteroides* and *Alistipes* in the colon were significantly higher than those in the ileum (*p* = 0, 0.001). The abundance of *Bifidobacterium* in the ileum was significantly higher than that in jejunum (*p* =0.0005).

### 3.4. Functional Prediction of Intestinal Bacterial Flora

PICRUST is a tool for predicting the function of microbial communities based on 16S rDNA sequences [23]. According to the abundance of functional information in each group at the tertiary levels, the top 20 functions terms of abundance were selected for cluster analysis and heat maps, where the color tends to be red to indicate higher abundance and vice versa (Figure 6).

The results show that there are four main types of functions, namely Cellular Processes (1.82%), Environmental Information Processing (9.29%), Genetic Information Processing (9.99%), and Metabolism (13.63%). The functions are mainly divided into two parts, one related to the intestinal environment adaptation, such as Two-component system, ABC transporter, Transcription factors, and the other related to cell reproduction, such as DNA replication proteins, DNA repair, and recombination proteins. The genetic information processing and metabolism of the large intestine are relatively active. In contrast to the large intestine, the abundance of cellular processes and environmental information processing in the small intestine is relatively high. In addition, the relative abundance of each type of function varied among intestinal segments, which was higher in the jejunum (9.41%), moderate in the cecum (8.26%) and colon (8.00%), and lower in the duodenum (7.55%).

## 4. Discussion

Before next-generation sequencing (NGS), the study of intestinal microorganisms was mainly carried out by purification and isolation techniques, and only a few microorganisms could be isolated at a time, which did not provide a comprehensive understanding of the composition and distribution of microorganisms in the animal intestine [24]. Furthermore, the interrelationships between microbial populations in different parts of the gut and the interactions between these populations [25]. With the advancement of technology, NGS has been widely used in the field of gut microbiology, providing a more direct and convenient method to analyze microbial community composition [26]. In animal gut microbial studies, most of the fecal samples are used to represent the microbial structure of the right dorsal colon to some extent, but not the proximal hindgut and the entire gut microbial community [27]. Therefore, sampling from different parts of the whole intestine can better reflect the composition and function of complex bacterial communities in the animal intestine [28]. In this study, we compared the microbial community structure and microbial composition in the contents of different parts of the intestinal tract of Hainan black goats. The results showed that there were differences in bacterial community composition, abundance, and diversity in different parts of the intestine of Hainan black goats.

According to the annotated results of OTUs and Alpha diversity analysis, the species richness and diversity of cecum and colon of Hainan black goats were higher than that of duodenum jejunum and ileum, which was different from the results of previous studies on the intestinal contents of Yimeng black goats [20], the reason might be that the breeds and feeding methods were different.

The dominant groups of intestinal microorganisms in ruminants play an important role in ecology and function. In this study, it was found that Firmicutes, Bacteroidota, and Pseudomonadota were the dominant phyla in the intestine of Hainan black goats, which was consistent with previous findings. Interestingly, in contrast to Yimeng black goat, Hainan black goat has a higher intestinal content of Bacteroidota in the intestine (except ileum), and this result is similar to sheep [20,29]. Therefore, we speculate that this result may be related to the breed and breeding environment. It can be seen that even for the same species, the composition of gut microbes might be different due to different environments. While the species richness and diversity are lower in the ileum, the relative abundance of Firmicutes is the highest in the ileum. Studies showed that Firmicutes have the ability to degrade cellulose, proteins, and carbohydrates, while the Bacteroidota can increase the utilization of nutrients by the host [30]. Gut-associated Bacteroidotas are mainly anaerobic polysaccharide-degraders, and its content is related to the intake of cellulose and polysaccharides of hosts [31]. Pseudomonadota plays an essential role in maintaining the structural stability and homeostasis of the intestinal flora and is a crucial indicator for determining the intestinal health of animals [32].

In contexts of the genus, the dominant groups in the gut of black goats mainly include Paeniclostridium, Ruminococcaceae_UCG-005, Bacteroides, Christensenellaceae_R-7_group, Rikenellaceae_RC9_ gut_group, Eubacterium coprostanoligenes _group, Prevotella_1. Some studies have shown that the gut microbiota composition of goats varied with age [12,19,33]. In general, Prevotella, Ruminococcus, and Bacteroides are the dominant groups in the gastrointestinal tract of adult goats. Additionally, Ruminococcaceae_UCG-005 and Christensenellaceae_R-7_group are dynamically distributed in the host intestine [19]. Prevotella and Ruminococcus are the dominant microorganisms in the rumen of adult ruminants, which mainly play an important role in degrading cellulose and polysaccharides [34]. In this study, we found that the relative abundance of Prevotella in the duodenum is higher than that in other parts of the intestine, which may be due to the connection between the duodenum and the rumen, and the food may flow into the duodenum with rumen fluid during ruminant digestion. Ruminococcus, especially Ruminococcaceae_ UCG-005, shows a gradual trend in the intestines of Hainan black goats. The relative abundance of Ruminococcus increased from the duodenum to the cecum and colon. The reason might be that the cecum is the main fermentation site in the intestinal tract of ruminants, and the food residue accumulates for a long time. In addition, we found that most of the microorganisms in the cecum were anaerobes, so the high relative abundance of Ruminococcus in the cecum may also be related to the anaerobic environment of the cecum. In general conditions, Ruminococcaceae are capable of degrading cellulose and starch, which has a beneficial effect on the host’s intestinal health [35]; remarkably, Ruminococcus could be a biomarker for goat health [36]. In this study, the relative abundance of Ruminococcaceae_UCG-005 was significantly higher in the cecum than in the duodenum, jejunum, and ileum (*p* < 0.05). As the common bacterium in the intestinal tract, Bacteroides have the ability to utilize polysaccharides and can improve nutrient utilization, accelerate the formation of intestinal mucosa, improve the host’s immune system, and maintain intestinal micro ecological balance [37]. In the present study, the relative abundance of Bacteroides was more abundant in the colon. The Christensenellaceae R-7 group was able to metabolize carbohydrates efficiently and produce acetate and butyric acid [38]. Research suggests that Christensenellaceae deficiency may be a cause of obesity [39]. Numerous studies have shown that not only inflammation or obesity, but Christensenellaceae may also be associated with other health problems in the host [38,39,40,41,42].

In the difference analysis of the microbial abundance of different intestinal segments, there were significant differences in the abundance of *Ruminococcaceae_UCG-005, Ruminiclostridium, Anaerofilum, Bacteroides, Alistipes,* and *Bifidobacterium*. *Anaerofilum* may be involved in the production of SCFAs in animals, which is positively correlated with the feed conversion ratio of chicken [43]. Generally, the abundance of *Anaerofilum* is 0.01%, and there is a risk of inflammation when the abundance is high [44]. *Alistipes* belongs to gram-negative bacteria and anaerobes in the intestinal environment [45]. *Bacteroides* has the function of degrading cellulose [46], and *Alistipes* is related to the formation of succinic acid [47]. Kim believes that there may be some interaction between Bacteroides and *Alistipes* [46]. Some studies have shown that *Alistipes* dynamic balance in the intestine is very important. *Alistipes* imbalance will lead to enteritis, which is considered as a biomarker for the diagnosis of human rectal cancer [48]. *Bifidobacterium* belongs to anaerobic bacteria, which can consume sugars in the intestinal tract and produce bacteriostatic substances [49] and has the effect of reducing lipids [46]. On the whole, the higher microbial abundance in the large intestine may be related to feed digestion and the intestinal environment. Most of the feed entering the large intestine is excreted out of the animal body as feces, which contain a large amount of undigested cellulose and organic acids [50]. Moreover, the longer residence time of feed in the large intestine is also conducive to the proliferation of these anaerobic microorganisms. PICRUSt showed that the adaptive functions are higher in the small intestine, such as Two−component system, ABC transporter, and transcription factors. It might be that the small intestine is the main part of nutrient absorption, and the frequent operation of the transport system, motor system, and even signal factors may lead to the strong expression of transcription factors in the small intestine. On the other hand, genetic replication and metabolism are more active in the caecum and colon, which may be related to the greater abundance of intestinal microorganisms in these intestines.

## 5. Conclusions

In summary, this study researched the gut microbial compositions of each intestinal segment of the Hainan black goat by collecting the intestinal contents of the duodenum, jejunum, ileum, cecum, and colon. The results revealed that the gut microbiotas are influenced by the intestinal location where they colonize in Hainan black goats. In Hainan black goat, the colon and caecum have richer and more complex gut bacteria than the duodenum, jejunum, and ileum, and the microbial composition in the cecum and colon was more similar. *Firmicutes, Bacteroidota,* and *Pseudomonadota* were the dominant phyla in the gut of Hainan black goats. The relative abundance of each type of function was higher in the jejunum, moderate in the cecum and colon, and lower in the duodenum. The present study is preliminary, observations and relationships of the microbiome to animal health and performance could be better observed by a greater understanding of individual variation, individual dietary habits, and connecting in more detail husbandry and diet metadata.

## Figures and Tables

**Figure 1 animals-12-03129-f001:**
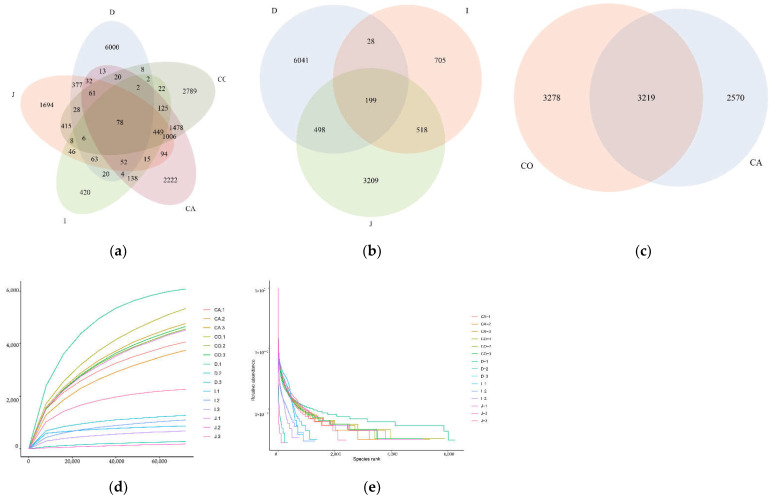
Venn diagram analysis and feasibility analysis based on the OUTs of each part. (**a**) The Venn diagram of each intestinal segment. (**b**) Venn diagram of various parts of the small intestine. (**c**) Venn diagram of various parts of the large intestine. (**d**) The species accumulation curve for each group. (**e**) The rank abundance curve for each group.

**Figure 2 animals-12-03129-f002:**
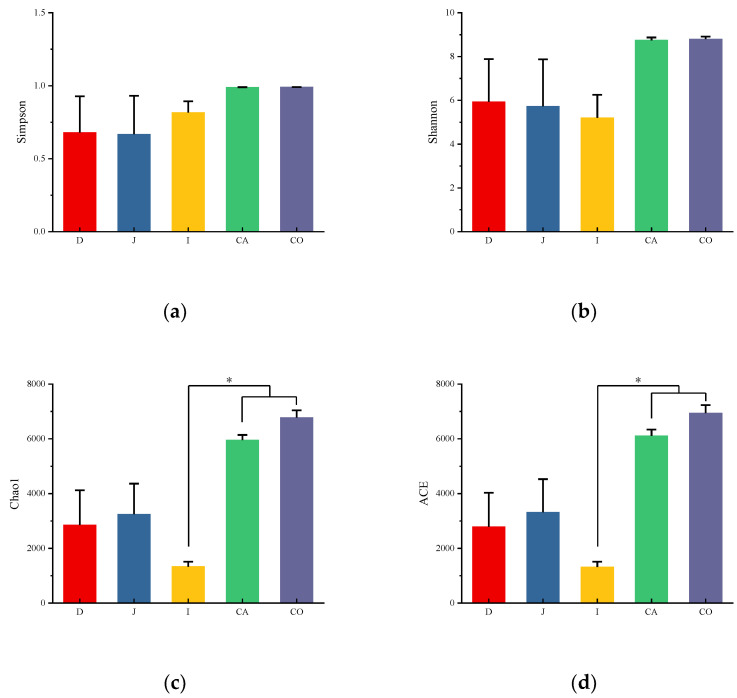
Alpha diversity in different intestinal segments of Hainan black goats. These show the Simpson index (**a**), Shannon index (**b**), Chao1 index (**c**), and ACE index (**d**) of each intestinal segment, respectively. * *p* < 0.05.

**Figure 3 animals-12-03129-f003:**
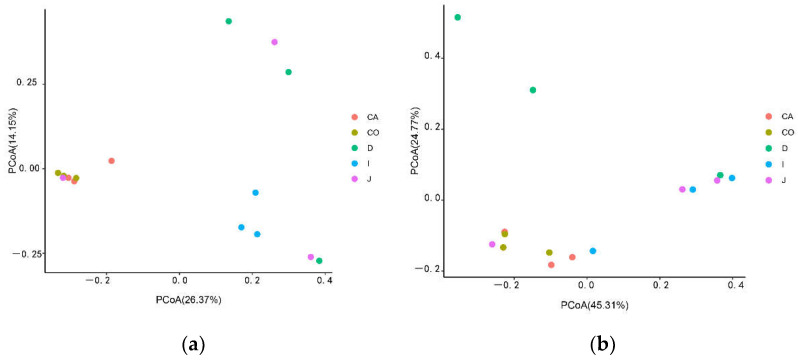
PCoA derived UniFarc distance map. (**a**) Shows the unweighted UniFrac distance. (**b**) Shows the weighted UniFrac distance. Each sample is represented by a point on the map. The difference in gut microbiota between samples is expressed as the distance between two points.

**Figure 4 animals-12-03129-f004:**
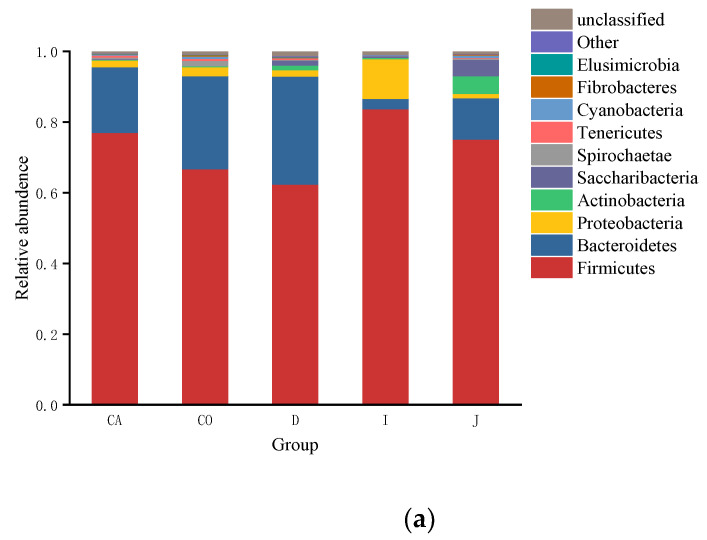
Relative abundance of microorganisms in each group. Both (**a**) and (**b**) show the relative abundance of intestinal microorganisms in each group in the contexts of phylum and genus.

**Figure 5 animals-12-03129-f005:**
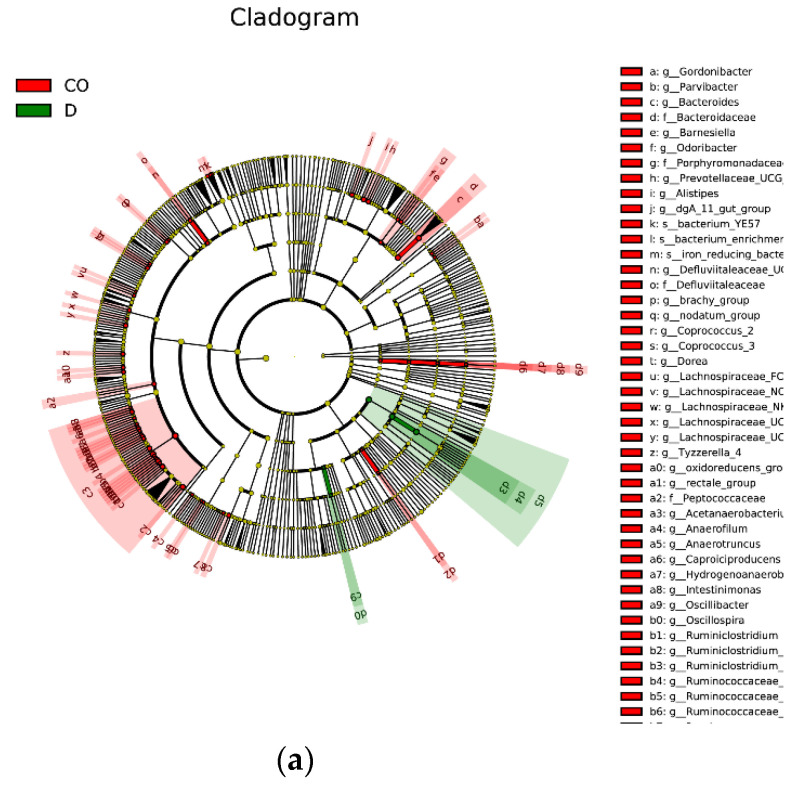
The evolutionary branching graph was obtained by LEfSe. In the evolutionary cladistic diagram, the circle of radiation from inside to outside represents the taxonomic level from phylum to genus (or species). Yellow means no significant difference. (**a**) LEFse difference analysis of duodenum (D) and colon (CO); (**b**) LEFse difference analysis of Ileum (I) and colon (CO); (**c**) LEFse difference analysis of Ileum (I) and Caecum (CA).

**Figure 6 animals-12-03129-f006:**
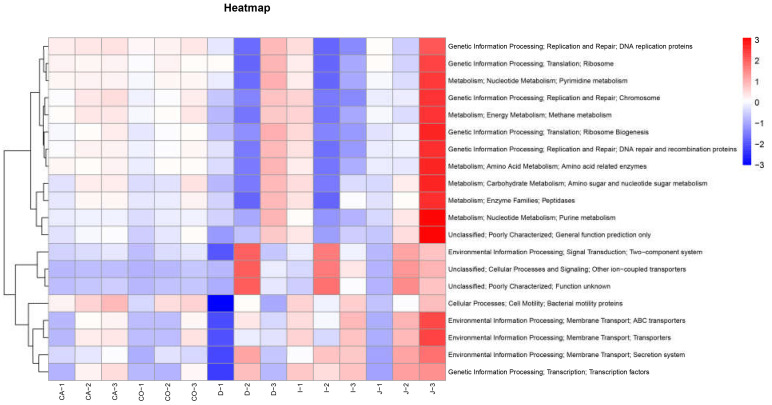
PICRUST annotates relative abundance cluster heat map among different samples of Hainan black goats.

**Table 1 animals-12-03129-t001:** Comparison of microbial abundance among different intestinal segments. Different letters represent significant differences. “ND” means not detected.

Taxa	D	J	I	CA	CO
Alistipes	0.0002409 ± 0.0002362 ^c^	0.0114919 ± 0.0114387 ^b^	0.0029477 ± 0.0009794 ^b^	0.0290308 ± 0.0039248 ^a^	0.030547 4± 0.0031616 ^a^
Anaerofilum	ND	0.0000254 ± 0.0000254 ^b^	ND	0.0000744 ± 0.0000065 ^bc^	0.000129 ± 0.0000468 ^a^
Bacteroides	0.0015784 ± 0.0015367 ^c^	0.0348161 ± 0.0347024 ^bc^	0.0063273 ± 0.0022585 ^c^	0.0402566 ± 0.01842 ^b^	0.0870625 ± 0.0059029 ^a^
Bifidobacterium	0.0000543 ± 0.0000543 ^a^	ND	0.0000112 ± 0.0000015 ^a^	ND	0.0000348 ± 0.0000244 ^a^
Ruminiclostridium	0.0000062 ± 0.0000062 ^b^	0.000189 ± 0.0001767 ^b^	0.0000265 ± 0.0000212 ^b^	0.0004042 ± 0.000026 ^a^	0.0005055 ± 0.0001536 ^a^
Ruminococcaceae_UCG-005	0.0016093 ± 0.0009648 ^c^	0.0368468 ± 0.0347901 ^bc^	0.0512074 ± 0.044556 ^bc^	0.1506268 ± 0.0100503 ^a^	0.1170258 ± 0.0102332 ^ab^
Ruminococcaceae_UCG-009	0.0000093 ± 0.0000093 ^d^	0.0013664 ± 0.0013626 ^c^	0.0046782 ± 0.0045223 ^b^	0.0097851 ± 0.0026403 ^a^	0.0057623 ± 0.0005336 ^b^

## Data Availability

All the raw data during the current study are available from the corresponding author.

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
