# Peer review of "Research on the Gut Microbiota of Hainan Black Goat"

_animals, 2022, doi:10.3390/ani12223129_

Round 1
Reviewer 1 Report (Previous Reviewer 2)
In the present study, the authors used HTS to investigate the microbial community structure, and function in different intestinal segments of Hainan black goats. they aimed to reveal differences in the community structure and species composition of the microbiota between the various intestinal segments and to identify the shared and unique bacterial species of gut microbiota in Hainan black goats.
- at the end of the abstract, you need to add, what your results indicate.
- In sample collection, please write the number of the collected samples and the characteristics of the animals that you selected.
- Why you selected the V3/V4. V4/V5 region is more conserved.
- add more details in the bioinformatic analysis section.
- Put the statistical analysis in a separate section and write it in detail.
- Please add more details/ citations in the discussion to compare with other studies on goat microbiota composition.
Author Response
Thank you for your comments. The statistical analysis section has been added. This revision explains the selection of primer V3V4 and V4V5. In addition, differential analysis of each intestinal segment was added to the manuscript. Please see the attachment and manuscript for details.
Reviewer 2 Report (New Reviewer)
A gut microbiome study was done on the Hainan black goat. I make several corrections and various comments that would improve this paper.
At least two other studies on the Hainan black goat are available. The authors should try to incorporate in the discussion and where relevant results the outcomes from a recent paper (Li et al. 2022) relevant paper
Li L, Li K, Bian Z, Chen Z, Li B, Cui K and Wang F (2022) Association between body weight and distal gut microbes in Hainan black goats at weaning age. Front. Microbiol. 13:951473. doi: 10.3389/fmicb.2022.951473
The authors also should do compositional based analysis including differential analysis comparing the intestinal segments. This will give results more accurate and potentially useful for future studies.
See this paper for the general approach – many other articles have used this approach since.
Gloor GB, Macklaim JM, Pawlowsky-Glahn V, Egozcue JJ. Microbiome Datasets Are Compositional: And This Is Not Optional. Front Microbiol. 2017 Nov 15;8:2224. doi: 10.3389/fmicb.2017.02224.
I assume the authors used PiCRUST? The authors ideally should use PiCRUST2 or Tax4Fun2 – these are updated programs and better for the functional inference analysis.
For future studies I strongly suggest higher sample numbers though there is cost and ethical implications.
The authors should be clear of any statistical approaches they use as well as appropriately stating (and using) p-values (see comment below).
Line 16 The Hainan black goat is a precious genetic resource in China….
Revise the following sentence
Line 17 However, updated research on topics concerning the gut microbiota of Hainan black goat are absent.
For example “However, research on the gut microbiota of Hainan black goat is only at an early stage.
Line 26 The Hainan black goat is a unique species in…
Line 32, line 157, line 209, line 211, Line 214 and elsewhere in the manuscript. When stating significance it would be best to give the actual number (or range of numbers) rather than just summarising it as p<0.05. Saying only p<0.05 does not give a reader an appreciation of how significant the relationships actually are.
Line 34 …Bacteroidetes and Proteobacteria were dominant in Hainan black goat gut samples.
Also update phylum names - Bacteroidetes is now Bacteroidota while Proteobacteria is now Pseudomonadota. See Oren & Garrity 2021.
Oren A, Garrity GM (2021). "Valid publication of the names of forty-two phyla of prokaryotes". Int J Syst Evol Microbiol. 71 (10): 5056.
Line 37 More accurately coprostanoligenes _group should be [Eubacterium] coprostanoligenes_group. (Eubacterium_R in GTDB https://gtdb.ecogenomic.org/searches?s=al&q=coprostanoligenes)
Lines 38-39 The PiCRUST information in the abstract is very vague and should be rewritten as a more well-defined outcome, otherwise please delete.
Line 46 …different microbial structures…
Lines 53-55 Correct sentence. It is broken up to make the meaning clearer.
Hainan black goat is mainly used for meat. The meat odor lacks a urine smell, the fat composition is uniform and not greasy, and has a high nutritional value. Hainan black goat meat is regarded as one of the "four famous foods" of Hainan [13,14].
Line 56 Again correct the sentence.
The meat protein content is more than 22.6% and is rich in most essential amino acids while fat and cholesterol content is less than 3% and 60 mg/kg, respectively.
Line 72 ….relationship between climate and microorganisms.
Would it not be more about animal physiology and climate since the bacteria are in a stable temperature in the gut (around 39-40 C)? Suggest modifying sentence to mention animal physiology, the effect on the gut microbiome is likely rather indirect.
For example “….relationship with climate and animal physiology and indirectly microbiomes”.
Line 88 …back to the laboratory on dry ice…
Why use dry ice? If shipping time to the laboratory was lengthy it would make sense perhaps. But dry ice would leave the intestinal samples frozen. Were they frozen when reaching the lab? This would make harder to do the downstream work. Chilling would have been enough since the microbes will not grow at all!
Line 110 The reads that have passed the primitively quality screening..
You mean I assume “The reads that have passed preliminary quality screening…”?
The word primitively is in the wrong context.
Line 113 High-quality sequences were clustered… Delete “The achieved”
Line 119 No details are actually given on what beta-diversity analysis was done. Methods should be mentioned here such as the ordination analysis used. The authors also should consider doing compositional based analysis including differential analysis comparing the intestinal segments.
See Gloor GB, Macklaim JM, Pawlowsky-Glahn V, Egozcue JJ. Microbiome Datasets Are Compositional: And This Is Not Optional. Front Microbiol. 2017 Nov 15;8:2224. doi: 10.3389/fmicb.2017.02224.
Lines 121-124 See comment on PICRUST above.
Lines 158-159 the species diversity within the ileum was lower than that found for duodenum and jejunum samples.
Line 160 This sentence is perplexing In contrast, Hainan black goat has higher species richness in the large intestine, which is the consistent with the Mongolian sheep…
Do you mean the goats have a diversity level similar to what has been observed in the Mongolian sheep? It is very unclear what you refer to here as well as the relevance. Also this reads as “discussion” not results. Please delete the sentence.
Line 192 …detected at a lower abundance.
Lines 231-233 Granted the functional inference is a rather descriptive and general analysis., the sentence here is especially vague to be useless. See also the comment for lines 38-39 in the abstract/summary. It is unclear what varies and by how much. Consider reanalysing using PICRUST2 or TAX4FUN2 – the results might become clearer.
For figure 5 only the functions are clustered you also need to cluster the samples (CA-1 etc.) to see if PICRUST etc. meaningful shows an actual relationship. To me cecum and colon seem similar while to duodenum is quite different. The other sections along with the duodenum seem rather variable/divergent. What is the control – the baseline being compared to for the heatmap analysis?
Line 267 ….higher intestinal content of Bacteroidetes (except the ileum), and this result is similar to that of sheep [19,20]
Line 274 Bacteroidetes is mostly anaerobes…
You can say instead. Gut-associated Bacteroidetes are mainly anaerobic polysaccharide-degraders and presence is related to…
Line 281 … composition of goats changes with age.
This fact is not surprising – this even happens in fish. Provide suitable references for goats related to this.
Line 298-229 and further below. How is the relationships shown in reference 34 relevant to goats (and other ruminants) specifically? It is fine making broad comparisons but one must be careful in doing so.
Lines 330-332 The final sentence is rather deflating – revise it to
The present study is preliminary, observations and relationships of the microbiome to animal health and performance could be better observed by greater understanding of individual variation, individual dietary habits and connecting in more detail husbandry and diet metadata.
Author Response
Thank you very much. Thanks a million. I don't know how to thank you.
Thank you very much for your valuable comments and suggestions on our manuscript.
We will collect as many samples as possible in future research work.
I added the intestinal segment difference analysis part in the manuscript to show the species abundance difference of different intestinal segments. I searched relevant materials and learned that PICUST2 is a new generation of function prediction software. But I am not familiar with it, so please forgive me for not changing it to PICRUST2.
In order to express the PICRST function prediction more clearly, I changed the scale color of the heat map and added PICUST data to the relevant position of the manuscript.
Please see the attachment and manuscript for details.

Round 2
Reviewer 1 Report (Previous Reviewer 2)
The authors addressed all the raised concerns. However, they still need English revision for typos.
Author Response
Thank you for reviewing my manuscript and giving me your review suggestions. I thank the reviewers for forgiving my poor English, and I have revised the English description in the yellow part of the manuscript
This manuscript is a resubmission of an earlier submission. The following is a list of the peer review reports and author responses from that submission.
Round 1
Reviewer 1 Report
I find it very difficult to understand what was the real purpose of the study and it is related to what was proposed on thesp Hainan goat. I was hoping to find some answers to why the Hainan goat is special and unique.
The authors completely ignored the fact the goat is a ruminant species and ignored the influence of ruminal bacterial flux to the duodenum.
The authors did not how and why the microbiota in the Intestine makes the Hainan goat a special one.
Reviewer 2 Report
This study aimed to reveal differences in the community structure and species composition of the microbiota between the different intestinal segments and to identify the shared and unique bacterial species of gut microbiota in Hainan black goats.
1- How many goats were included in this study and their sex and declare if they have been raised in the same area?
2- You need to mention the environmental condition, where the animals were raised, the type of food, and any other conditions.
3- DNA was extracted from which samples, feces or tissue?
4- Why did you select region V3/V4. why did not you select V4/V5 which is more conservative?
5-What about the beta diversity?
6- put the statistical analysis in a separate section.
7- Figure 4: in Y-axis: relative abundance in %?
8- Why do you think there is a difference between D-2 and J-4 and the other samples.
9- You describe the abundance of each bacteria in each part but you still need to add an indication for your results. For example, if bacteria x increases, is this related to mean production, weight gain, etc?